# Weekly Variations of Well-Being and Interactions with Training and Match Intensities: A Descriptive Case Study in Youth Male Soccer Players

**DOI:** 10.3390/ijerph19052935

**Published:** 2022-03-02

**Authors:** Ana Filipa Silva, Rafael Oliveira, Stefania Cataldi, Filipe Manuel Clemente, Francesca Latino, Georgian Badicu, Gianpiero Greco, César Leão, Valerio Bonavolontà, Francesco Fischetti

**Affiliations:** 1Escola Superior Desporto e Lazer, Instituto Politécnico de Viana do Castelo, Rua Escola Industrial e Comercial de Nun’Álvares, 4900-347 Viana do Castelo, Portugal; anafilsilva@gmail.com (A.F.S.); filipe.clemente5@gmail.com (F.M.C.); cleao@esdl.ipvc.pt (C.L.); 2Research Center in Sports Performance, Recreation, Innovation and Technology (SPRINT), 4960-320 Melgaço, Portugal; 3The Research Centre in Sports Sciences, Health Sciences and Human Development (CIDESD), 5001-801 Vila Real, Portugal; rafaeloliveira@esdrm.ipsantarem.pt; 4Sports Science School of Rio Maior-Polytechnic Institute of Santarém, 2040-413 Rio Maior, Portugal; 5Life Quality Research Centre, 2040-413 Rio Maior, Portugal; 6Department of Basic Medical Sciences, Neuroscience and Sense Organs, University of Study of Bari, 70124 Bari, Italy; stefania.cataldi@uniba.it (S.C.); francesca.latino@uniba.it (F.L.); gianpierogreco.phd@yahoo.com (G.G.); francesco.fischetti@uniba.it (F.F.); 7Instituto de Telecomunicações, Delegação da Covilhã, 1049-001 Lisboa, Portugal; 8Department of Physical Education and Special Motricity, University Transilvania of Brasov, 500068 Brasov, Romania; georgian.badicu@unitbv.ro

**Keywords:** football, wellness, exercise, fatigue, sleep, readiness, muscle soreness, perceived exertion, load

## Abstract

The aim of this study was two-fold: (i) analyze the weekly variations of well-being and training/match intensity measures in youth soccer players, and (ii) test relations between well-being and training intensity outcomes. The study followed a descriptive case study design. Twenty-seven under-17 male soccer players were monitored for well-being and training intensity parameters over seventeen consecutive weeks. An adjusted version of the Hooper questionnaire was used to monitor the perceptive sleep quality, readiness, fatigue, and delayed onset muscle soreness (DOMS) early in the morning. The CR-10 Borg’s scale was also used for monitoring the rate of perceived exertion (RPE) of players after training sessions. Repeated-measures analysis of variance was executed to test the between-week variations of both well-being and training intensity outcomes. Moreover, Pearson product moment correlation was used to test the relations between well-being and training intensity outcomes. Repeated measures ANOVA revealed significant differences between weeks in the sleep quality (*F* = 0.422; *p* < 0.001; $\eta_{p}^{2}$ = 0.140), readiness (*F* = 0.8.734; *p* < 0.001; $\eta_{p}^{2}$ = 0.251), fatigue (*F* = 4.484; *p* < 0.001; $\eta_{p}^{2}$ = 0.147), DOMS (*F* = 3.775; *p* = 0.001; $\eta_{p}^{2}$ = 0.127), RPE (*F* = 7.301; *p* < 0.001; $\eta_{p}^{2}$ = 0.219), and session-RPE (*F* = 17.708; *p* < 0.001; $\eta_{p}^{2}$ = 0.405). Correlations between well-being and training intensity outcomes in the same week revealed moderate correlations between fatigue and session-RPE (*r* = 0.325). As conclusions, it was found that well-being and training intensity fluctuates over the season, while well-being outcomes seems to be related with training intensity, although with a small magnitude.

## 1. Introduction

Managing the training process while monitoring the impact of training stimulus on the soccer players makes up part of the tasks of coaches and practitioners [1,2]. Currently, applying an athlete’s monitoring cycle in which training demands is a well-implemented practice in soccer clubs. As an example, a study summarizing results of a survey performed at twenty-eight European soccer clubs [3] revealed that 100% of the clubs use monitoring processes, most of them monitoring locomotor demands and half of them additionally monitoring psychophysiological demands. Additionally, in a survey conducted on 84 coaches and 88 practitioners it was revealed that coaches and practitioners sometimes adjust training sessions based on previous training intensity monitoring, and that training intensity reports are often provided to coaches [4]. Thus, monitoring locomotor and psychophysiological demands imposed by training and/or matches is a usual practice in both adults and youth categories [3,5].

While monitoring training demands is current practice, other factors should be considered for properly understanding the impact of training and match stimulus on the players’ responses [6]. Thus, an athletes’ monitoring cycle is proposed as a recommended practice to implement in any training scenario [7]. The athletes’ monitoring cycle consists of monitoring training demands (e.g., locomotor/mechanical and psychophysiological), as well as the well-being and the readiness of players [7]. In this conceptual framework, perceptual well-being is related to training intensity, namely representing the way players are coping with training demands [7]. The authors of this concept [7] also suggest that poor perceptual well-being and high training demands should adhere to an adjustment in the training dose, while a high training demand followed by a good level of perceptual well-being is a signal to continue the training process.

Descriptive studies have been tried to test this interaction between training intensity and well-being outcomes in soccer, covering adults [8,9,10], and youth [11,12]. While direct relations between well-being outcomes (e.g., sleep quality, delayed onset muscle soreness (DOMS) mood, fatigue, stress) and training intensity (e.g., rate of perceived exertion, RPE) have revealed small-to-moderate magnitudes of correlation [13], specific original studies have been suggesting large magnitudes of correlations between well-being outcomes and some measures that identify accumulated training intensities and variability of these demands [11]. Possibly, fluctuations over the season can be a cause of that.

Seasonal variations of training intensities [14] and well-being outcomes [15] in soccer players have been described. In youth, well-being scores seem to be more stable in the middle of the season, while in early and ending phases of the season presents greater variability [15]. Interestingly, also in youth soccer players, significantly greater accumulated training demands were found in the middle of the season [11].

Although well-established psychometric instruments such as CR-10 Borg’s scale have been confirmed for their validity, reliability, and sensitivity [16], as well as well-being questionnaires as proposed by Hooper and colleagues [17], there are some factors that can influence the direction and magnitude of correlations between well-being and training intensity. For example, relationships between well-being outcomes and training intensity in the same day can be different to testing well-being and training intensity outcomes considering days of difference. This has not been reported, and it could be interesting to understand the possible delayed effects of accumulated training demands or accumulated poor well-being reports in the following training process. Possibly, better identification of such relationships may provide useful insight to coaches and parents for being attentive to some signals in players [18,19].

Considering the above-mentioned gap in the current research, it is important to describe the variations of well-being and training intensity outcomes and particularly inspect the relations between these outcomes with special attention to the effects of previously accumulated training intensity or accumulated well-being scores on the variations of the other parameters. Thus, the purpose of this study was two-fold: (i) analyze the weekly variations of well-being measures in youth soccer players, and (ii) test relations between well-being and training intensity outcomes.

## 2. Materials and Methods

### 2.1. Study Design

The study followed a descriptive case study design.

### 2.2. Setting

The observational period occurred between 29 July 2021 and 17 November 2021. Seventeen consecutive weeks were observed, including a total of 64 training sessions and 19 matches. The details about the observed period can be observed in Table 1. Over the period, the players were asked to fill out a wellness questionnaire (adjusted version of the Hooper questionnaire) and to rate the perceived exertion (RPE) regarding the effort associated with the training intensity. Moreover, the duration of the training sessions and/or matches was registered for further data treatment. Players only registered wellness scores in the same days in which the training and/or match occurred. The wellness scores were provided before the training started, while the RPE was scored between 20 and 30 minutes after the end of the training session and/or match. Typically, training sessions were structured in a warm-up, followed by analytic exercises focusing on the conditioning of players (e.g., aerobic, anaerobic, speed, or change-of-direction) and a period of exercise with small-sided games and positioning games. After that, a short period of 11 vs. 11 between players and a period of cool-down was implemented.

### 2.3. Participants

Convenience sampling was used in the current study. The players were recruited from the same team. Twenty-seven male soccer players (age: 16.3 ± 0.3 years; height: 1.8 ± 0.1 m; body mass: 67.7 ± 7.4 kg; body mass index: 22.1 ± 0.9 kg/m^2^) voluntarily participated in the observational period. The following eligibility criteria was considered for including players in the data treatment: (i) reported wellness and RPE scores every time they were part of training sessions and/or matches; (ii) participate in >90% of the training sessions occurring in the period of observation; (iii) participate in at least 50% of the matches occurring in the observational period; (iv) not exceed more than one week in missing data. The study design and protocol were preliminarily explained and detailed to the players and their parents. After being informed about the risks and benefits, they signed a free consent. The study has followed the ethical standards for the study in humans, in accordance with the Declaration of Helsinki.

### 2.4. Well-Being Questionnaire

An adjusted version from the proposed Hooper questionnaire [17] was used. An ordinal 10-point scale was used. The score and verbal anchors can be found in Table 2. The questionnaire was preliminarily introduced to the athletes in the previous two weeks before starting the observational period, aiming to familiarize them. The scores were provided before each training session and/or match, about thirty minutes before. The scores were provided individually, and the answers were registered by the observer in a database. The main outcomes extracted for further data treatment were the scores in sleep quality, readiness, fatigue, and delayed onset muscle soreness (DOMS) categories analyzed by the questionnaire.

### 2.5. Training and Match Intensity

The training intensity was monitored using the CR-10 Borg’s scale [20]. The score of the scale varies between 0 (nothing at all) and 10 (extremely strong) to the question “how intense was your training session?”. The scores can be provided from 0.5 to 0.5. The CR-10 Borg’s scale was applied between 20 to 30 minutes from the end of training session and/or match. The scores were provided individually, and the observer collected the information in a database. The players were previously familiarized with the scale. The score provided was used as RPE outcome for further statistical treatment. Additionally, the session-RPE [21] per each training session and/or match was calculated as follows: CR10 Borg’s scale score × time of the entire session (minutes). The session-RPE was also used as main outcome of the current research.

### 2.6. Statistical Procedures

The descriptive statistics are presented in the form of average and standard deviation. Normality and homogeneity of the sample was tested using the Shapiro–Wilk test and Levene’s test, respectively. After confirmation of the normality and homogeneity assumptions (*p* > 0.05), a repeated measures ANOVA was conducted to analyze the variations of wellness and training intensity scores over the seventeen weeks. The Bonferroni’s post hoc test was used to test the pairwise comparisons. Partial eta squared was executed to determine the effect size of analysis of variation. Aiming to analyze the relations between wellness and training intensity outcomes, a Pearson product moment correlation test was executed. Average and confidence intervals of correlation coefficient (*r*) were presented. Magnitude of correlations were classified based on the following thresholds [22]: (0.0–0.1) trivial; (0.1–0.3) small; (0.3–0.5) moderate; (0.5–0.7) large; (0.7–0.9) very large; (0.90-1.00) nearly perfect. All the statistical procedures were executed in the SPSS (version 28.0.0.0, IBM, Chicago, IL, USA) for a *p* < 0.05.

## 3. Results

Descriptive statistics of well-being and training intensity outcomes can be found in Table 3. Moreover, a graphical representation of the outcomes over the period of observation can be observed in Figure 1.

Repeated measures ANOVA revealed significant differences between weeks in the sleep quality (*F* = 0.4216; *p* < 0.001; $\eta_{p}^{2}$ = 0.140), readiness (*F* = 8.734; *p* < 0.001; $\eta_{p}^{2}$. = 0.251), fatigue (*F* = 4.484; *p* < 0.001; $\eta_{p}^{2}$ = 0.147), DOMS (*F* = 3.775; *p* = 0.001; $\eta_{p}^{2}$ = 0.127), RPE (*F* = 7.301; *p* < 0.001; $\eta_{p}^{2}$ = 0.219), and session-RPE (*F* = 17.708; *p* < 0.001; $\eta_{p}^{2}$ = 0.405).

The sleep quality score on week 2 was significantly smaller than in week 6 (−0.8 A.U.; *p* = 0.013). Regarding readiness scores, week 1 had significant smaller values than weeks 11 (−0.5 A.U.; *p* = 0.028), 12 (−0.6 A.U.; *p* = 0.002), and 14 (−0.5 A.U.; *p* = 0.024). Week 2 also had significantly smaller readiness scores than weeks 12 (−0.7 A.U.; *p* = 0.008) and 13 (−0.6 A.U.; *p* = 0.042). Week 3 presented significantly smaller readiness scores than weeks 11 (−0.5 A.U.; *p* = 0.031), 12 (−0.6 A.U.; *p* < 0.001), 13 (−0.5 A.U.; *p* = 0.025), and 14 (−0.6 A.U.; *p* = 0.025). Week 4 presented significantly smaller readiness scores than weeks 6 (−0.4 A.U.; *p* = 0.005), 9 (−0.4 A.U.; *p* = 0.043), 11 (−0.6 A.U.; *p* = 0.010), 12 (−0.7 A.U.; *p* < 0.001), 13 (−0.6 A.U.; *p* < 0.001), and 14 (−0.6 A.U.; *p* = 0.001). Week 5 presented significantly smaller values of readiness scores than week 12 (−1.1 A.U.; *p* = 0.043).

Considering fatigue, week 11 presented significantly smaller values than week 1 (−0.7 A.U.; *p* = 0.003), week 2 (−0.9 A.U.; *p* = 0.003), week 4 (−1.0 A.U.; *p* < 0.001), and week 8 (−0.7 A.U.; *p* = 0.009). Week 4 had significantly greater fatigue scores than week 6 (+0.8 A.U.; *p* = 0.003), week 9 (+0.6 A.U.; *p* = 0.032), week 10 (+0.7 A.U.; *p* = 0.012), and week 14 (+0.7 A.U.; *p* = 0.029).

Week 1 had significantly greater DOMS scores than week 5 (+0.8 A.U.; *p* = 0.023), week 6 (+0.9 A.U.; *p* < 0.001), week 11 (+1.3 A.U.; *p* < 0.001), week 13 (+1.0 A.U.; *p* = 0.007), and week 14 (+0.9 A.U.; *p* = 0.016). Week 11 had significantly smaller DOMS scores than week 8 (−0.9 A.U.; *p* = 0.002), week 12 (−0.7 A.U.; *p* = 0.009), and week 16 (−0.8 A.U.; *p* = 0.022).

Regarding RPE scores, Week 1 had significantly greater values than week 7 (+0.5 A.U.; *p* = 0.025), week 12 (+0.7 A.U.; *p* < 0.001), week 15 (+0.5 A.U.; *p* = 0.022), and week 17 (+0.9 A.U.; *p* = 0.007). Significantly greater RPE scores were found in week 4 in comparison to weeks 7 (+0.7 A.U.; *p* < 0.001), 10 (+0.5 A.U.; *p* = 0.020), 11 (+0.9 A.U.; *p* = 0.018), 12 (+1.0 A.U.; *p* < 0.001), 15 (+0.7 A.U.; *p* < 0.001), 16 (+0.6 A.U.; *p* < 0.001), and 17 (+1.3 A.U.; *p* < 0.001). Significantly greater RPE scores were found in week 5 than in weeks 7 (+0.6 A.U.; *p* = 0.008), 12 (+0.9 A.U.; *p* < 0.001), 15 (+0.6 A.U.; *p* = 0.010), and 17 (+1.2 A.U.; *p* = 0.002). Week 7 had significantly smaller RPE scores than week 13 (−0.7 A.U.; *p* = 0.004). Significantly greater RPE scores were found at week 8 in comparison to weeks 12 (+0.7 A.U.; *p* < 0.001) and 17 (+1.0 A.U.; *p* = 0.031). Significantly greater RPE scores were found at week 10 than in week 12 (+0.5 A.U.; *p* = 0.009). Significantly smaller RPE scores were found at week 12 than in weeks 13 (−1.0 A.U.; *p* < 0.001) and 16 (−0.4 A.U.; *p* = 0.007). Week 13 had significantly greater RPE scores than weeks 15 (+0.7 A.U.; *p* = 0.002), 16 (+0.6 A.U.; *p* = 0.023), and 17 (+1.3 A.U.; *p* < 0.001).

Regarding the session-RPE, week 1 presented significantly greater values than weeks 11 (+114.5 A.U.; *p* = 0.026), 12 (+126.4 A.U.; *p* < 0.001), and 17 (+199.0 A.U.; *p* < 0.001), while presented significantly smaller than week 4 (−112.7 A.U.; *p* = 0.003). Week 2 presented significantly greater session-RPE scores than weeks 6 (+141.3 A.U.; *p* = 0.019), 10 (+114.3 A.U.; *p* = 0.046), 11 (+147.3 A.U.; *p* = 0.011), 12 (+115.0 A.U.; *p* < 0.001), 14 (+115.0 A.U.; *p* < 0.001), 16 (+117.4 A.U.; *p* = 0.012), and 17 (+231.8 A.U.; *p* < 0.001). Week 3 had significantly greater session-RPE scores than weeks 11 (+138.0 A.U.; *p* = 0.017), 12 (+149.9 A.U.; *p* < 0.001), 14 (+105.7 A.U.; *p* = 0.008), 16 (+108.1 A.U.; *p* = 0.003), and 17 (+222.5 A.U.; *p* < 0.001). Week 4 had significantly greater session-RPE scores than weeks 6 (+221.2 A.U.; *p* < 0.001), 7 (+164.2 A.U.; *p* < 0.001), 8 (+139.4 A.U.; *p* < 0.001), 9 (+144.2 A.U.; *p* < 0.001), 10 (+194.2 A.U.; *p* < 0.001), 11 (+227.2 A.U.; *p* < 0.001), 12 (+239.1p A.U.; *p* < 0.001), 13 (+175.7 A.U.; *p* < 0.001), 14 (+194.9 A.U.; *p* < 0.001), 15 (+177.415 A.U.; *p* < 0.001), 16 (+197.3 A.U.; *p* < 0.001), and 17 (+311.7 A.U.; *p* < 0.001). Week 5 had significantly greater session-RPE scores than weeks 12 (+139.4 A.U.; *p* = 0.007), 14 (+95.2 A.U.; *p* = 0.039), 16 (+97.6 A.U.; *p* = 0.043), and 17 (+212.0 A.U.; *p* < 0.001). Week 7 had significantly greater session-RPE than week 17 (+147.5; *p* < 0.001). Significantly greater session-RPE was found in week 8 in comparison to weeks 11 (+99.7; *p* = 0.029) and 17 (+172.3; *p* < 0.001). Week 17 had significantly smaller session-RPE than weeks 9 (−167.5 A.U.; *p* < 0.001), 10 (−117.5 A.U.; *p* < 0.001), 11 (–84.5 A.U.; *p* = 0.023), 13 (−136.0 A.U.; *p* = 0.023), 14 (−116.8 A.U.; *p* < 0.001), 15 (−134,.3 A.U.; *p* < 0.001), and 16 (−114.4 A.U.; *p* < 0.001).

Correlations between well-being and training intensity outcomes scored in the same week are presented in Table 4. Moderate correlations were found between fatigue and session-RPE (*r* = 0.325). Small magnitudes of correlation were found between session-RPE and sleep (*r* = −0.119), readiness (*r* = −0.235), and DOMS (*r* = 0.161). Small magnitudes of correlation were found between RPE (*r* = 0.170) and fatigue and DOMS (*r* = 0.111).

Table 5 presents the correlations between well-being outcomes and the training intensities reported in the week immediately following the well-being reports. Small magnitudes of correlation were found between session-RPE and readiness (*r* = −0.115) and fatigue (*r* = 0.262). Small magnitudes of correlation were found between RPE and fatigue (*r* = 0.164) and DOMS (*r* = 0.102).

Correlation coefficients between training intensity and well-being outcomes reported the week after the training intensity reports can be found in Table 6. Small magnitudes of correlation were found between RPE and readiness (*r* = −0.135), fatigue (*r* = 0.202), and DOMS (*r* = 0.122). Similarly, small magnitudes of correlation were found between session-RPE and readiness (*r* = −0.167), fatigue (*r* = 0.282) and DOMS (*r* = 0.134).

## 4. Discussion

The aims of this study were to analyze the variations of well-being and intensity measures across 17 weeks in youth soccer players and to test associations between well-being and training intensity measures. Regarding the first aim, several significant differences between weeks for all well-being measures and training intensity were found.

Specifically, sleep quality was reported as good or higher for all weeks, and overall it seems that weeks with two matches reported higher values of sleep quality. This finding seems to be in line with previous studies that found that high-intensity training sessions performed in the evenings for young soccer players [23], or matches for professional soccer players, had no impact on sleep quality [9,24].

Readiness showed a tendency of higher values from week 6 forward. It seems that weeks with more matches cause a perception of higher readiness. Following the same line, fatigue and DOMS perceptions where higher values occurred in the first weeks and from week 6 forward, a tendency to lower the values occurred. Intensity measures of RPE and session-RPE seem to be in line with well-being measures, although there were some variations as well after week 6; a tendency in lowering the RPE and session-RPE values was observed until the last week analyzed.

Following a previous study, the well-being results seem to be in line, although different approaches for data analysis had been used [11]. In the Nobari et al. study, weekly accumulated data was used instead of weekly average data and the original Hooper index was used [11], but the results seem to be aligned. Other studies found lower values during mid-season (weeks 14 to 31) for sleep quality, DOMS, and fatigue than in earlier seasons (weeks 6 to 13) [12]. Although our study presents a different design and only 17 weeks in analysis, we would speculate different results because our data seems to support that the weeks with higher number of matches show a tendency to increase the well-being perception and to reduce the intensity. Indeed, this was in opposition to a previous study conducted with professional soccer players where weeks with two matches showed higher values of fatigue and DOMS than weeks with only one match [9].

Regarding intensity, previous studies also showed higher values from week 6 forward when compared to the results of the present study [11,12,25], but there was one study that showed higher values in the first month that tended to be reduced in the following two months [26] which seems to be in line with the present study. Despite the differences between studies, the RPE and session-RPE values found in this study seems to overcome the range values found in a recent systematic review conducted in young soccer players (RPE = 2.3–6.3 A.U.; session-RPE = 156–394 A.U.) [27].

From the second aim of this study, there was a moderate correlation between fatigue and session-RPE and small correlations between session-RPE and sleep, readiness, and DOMS; RPE and DOMS in the same week. The correlation between fatigue and session-RPE was also found in another study that used weekly accumulated data [12]. In fact, that study found correlations between session-RPE and DOMS and fatigue [12]. Another study in young soccer players also showed that fatigue, DOMS, and sleep were largely related to session-RPE [11]. In professional soccer players, session-RPE also displayed moderate correlations with fatigue and DOMS [28,29]. The previous correlations seem to support the findings of the present study.

Despite the fact that some differences exist, it seems that with higher intensity, higher levels of fatigue and DOMS occur, while at the same time higher levels of intensity seem to be associated with better readiness and sleep quality. This was also observed in our analysis when readiness and fatigue values were associated with both RPE and session-RPE of the week after. Furthermore, both RPE and session-RPE also showed associations with readiness, fatigue, and DOMS. It seems that this was the first study that conducted this type of analysis. Therefore, future studies should consider it to amplify knowledge in this field.

As mentioned in the beginning of this discussion, and despite the correlation shown, our data revealed that weeks with two matches tended to show better well-being and lower intensity. However, it important to highlight that the number of matches was not considered in the correlation analysis, which is required for future studies.

The present study presents some limitations, namely: the small sample size that came from only one team; an analysis of 17 weeks and not the entire season; the lack of locomotor measures (e.g., high-speed running, sprint, and accelerations) that could amplify the present results; and the lack of dietary control and supplementation. Finally, an intra-individual analysis considering the interaction between locomotor demands, playing position, physical fitness, and lifestyle was not analyzed and should be performed in future research aiming to explain the causes for variations. Therefore, future studies should avoid previous limitations and use: larger sample sizes and full-season analysis and external load measures. In addition, other contextual variables such as match results could influence the results and should be considered for future studies as previously suggested. For instance, a match win showed to provide better sleep quality when compared with a draw or a loss [30]. In the same line, match location should be taken in consideration in future analysis because it has been shown that away matches that required longer distance of travelling showed sleep/wake behavior impairment [31]. Moreover, analysis of dietary intake and supplementation should be considered, namely trying to establish relationships with wellness and coping with training demands.

Moreover, some studies have shown the importance of playing positions due to the different physical and physiological demands [27] and several variations in well-being [15], as well as playing status (starters and non-starters) that reflect differences across the season in young soccer players [26]. For that reason, they should be considered in future research.

Lastly, similar designs should be replicated not only for young soccer players, but also for professional elite men and women players. Additionally, future studies should analyze the influence of congested periods (weeks with two or more matches) compared with regular weeks (weeks with only one match).

Nonetheless, this study should be considered by coaches and their staff to acknowledge the importance of internal intensity and wellbeing measures such as readiness, sleep quality, fatigue, and DOMS variables as a mandatory daily task.

## 5. Conclusions

This study showed that well-being and training intensity fluctuates over the weeks. In addition, well-being measures seem to be related to training intensity, although with a small magnitude (only a moderate correlation was found between session-RPE and fatigue). Even so, this study showed a tendency of lower internal intensity and better well-being in the weeks with two matches.

## Figures and Tables

**Figure 1 ijerph-19-02935-f001:**
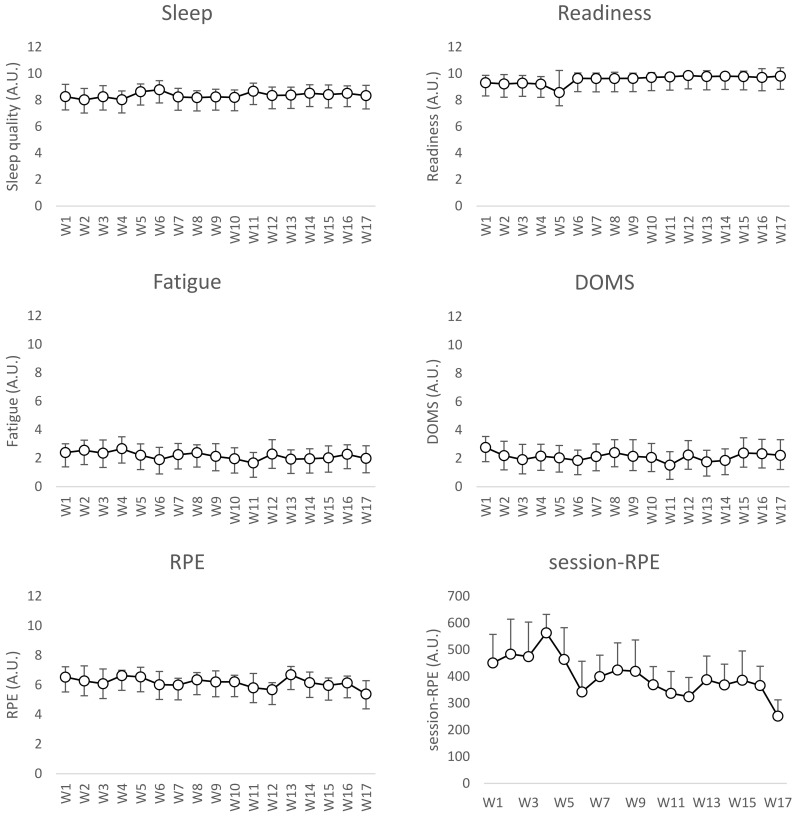
Weekly average values of sleep, readiness, fatigue, delayed onset muscle soreness (DOMS), CR10 Borg’s scale (RPE: rate of perceived exertion), and session-RPE. W: week; A.U.: arbitrary units.

**Table 1 ijerph-19-02935-t001:** Timeline of the study.

Week	Start Date	End Date	Sessions (N)	Matches (N)
Week 1	29 July 2022	1 August 2021	3	0
Week 2	3 August 2021	8 August 2021	4	1
Week 3	10 August 2021	14 August 2021	4	1
Week 4	17 August 2021	20 August 2021	4	0
Week 5	24 August 2021	28 August 2021	4	1
Week 6	30 August 2021	5 September 2021	3	2
Week 7	6 September 2021	12 September 2021	5	0
Week 8	13 September 2021	18 September 2021	4	1
Week 9	20 September 2021	26 September 2021	4	1
Week 10	27 September 2021	3 October 2021	4	2
Week 11	4 October 2021	10 October 2021	3	2
Week 12	11 October 2021	16 October 2021	3	2
Week 13	18 October 2021	24 October 2021	4	2
Week 14	25 October 2021	31 October 2021	4	2
Week 15	1 November 2021	6 November 2021	5	0
Week 16	8 November 2021	14 November 2021	4	2
Week 17	15 November 2021	17 November 2021	2	0

**Table 2 ijerph-19-02935-t002:** Wellness questionnaire (score and verbal anchors) used in the current study.

Score	Sleep Quality	Readiness	Fatigue	DOMS
10	Excellent	Totally available	Tired/Exhaustion	Extremely sore
9				
8	Good	Available	Very high	Very sore
7				
6				Sore
5	Moderate	Moderate	Moderate	Moderate
4				
3	Bad	Little available		
2			Light	Light
1	Very bad	Very little available		
0	No sleep	Nothing available	None	None

DOMS: delayed onset muscle soreness.

**Table 3 ijerph-19-02935-t003:** Descriptive statistics (mean ± standard-deviation) of well-being and training/match intensity outcomes over the observed period.

Week	Sleep Quality (A.U.)	Readiness (A.U.)	Fatigue (A.U.)	DOMS (A.U.)	RPE (A.U.)	Session-RPE (A.U.)
W1	8.3 ± 0.9	9.3 ± 0.5 ^w11,w12,w14^	2.4 ± 0.6 ^w11^	2.8 ± 0.8 ^w5,w6,w11,w13,w14^	6.5 ± 0.7 ^w7,w12,w15,w17^	450.7 ± 106.9 ^w4,w11,w12,w17^
W2	8.0 ± 0.9 ^w6^	9.2 ± 0.7 ^w12,w13^	2.6 ± 0.7 ^w11^	2.2 ± 1.0	6.3 ± 1.0	483.5 ± 130.6 ^w6,w10,w11,w12,w14,w16,w17^
W3	8.3 ± 0.8	9.3 ± 0.6 ^w11,w12,w13,w14^	2.4 ± 0.9	1.9 ± 1.1	6.1 ± 1.0	474.2 ± 129.3 ^w11,w12,w14,w16,w17^
W4	8.0 ± 0.7	9.2 ± 0.6 ^w6,w9,w11,w12,w13,w14^	2.7 ± 0.8 ^w6,w9,w10,w11,w14^	2.2 ± 0.8	6.7 ± 0.4 ^w7,w10,w11,w12,w15,w16,w17^	563.4 ± 69.1 ^w1,w6-w17^
W5	8.6 ± 0.6	8.6 ± 1.7 ^w12^	2.2 ± 0.8	2.0 ± 0.9	6.6 ± 0.7 ^w7,w12,w15,w17^	463.7 ± 118.8 ^w4,w11,w12,w14,w16,w17^
W6	8.8 ± 0.7 ^w2^	9.6 ± 0.4 ^w4^	1.9 ± 0.9 ^w4^	1.9 ± 0.7	6.0 ± 0.9	342.2 ± 114.6 ^w2,w4^
W7	8.2 ± 0.6	9.6 ± 0.4	2.2 ± 0.8	2.1 ± 0.9	6.0 ± 0.5 ^w1,w4,w5,w13^	399.2 ± 80.6 ^w4,w17^
W8	8.2 ± 0.5	9.6 ± 0.5	2.4 ± 0.6 ^w11^	2.4 ± 0.9 ^w11^	6.4 ± 0.5 ^w12,w17^	424.0 ± 117.4 ^w4,w11,w17^
W9	8.2 ± 0.6	9.6 ± 0.4 ^w4^	2.1 ± 0.9 ^w4^	2.1 ± 1.2	6.2 ± 0.7	419.2 ± 117.4 ^w4,w17^
W10	8.2 ± 0.6	9.7 ± 0.4	2.0 ± 0.8 ^w4^	2.1 ± 1.0	6.2 ± 0.4 ^w4,w12^	369.2 ± 67.9 ^w2,w4,w17^
W11	8.7 ± 0.6	9.8 ± 0.3 ^w1,w3,w4^	1.7 ± 0.7 ^w1,w2,w4,w8^	1.5 ± 1.0 ^w8,w12,w16^	5.8 ± 1.0 ^w4^	336.2 ± 81.8 ^w1,w2,w3,w4,w5,w8,w17^
W12	8.4 ± 0.6	9.9 ± 0.2 ^w1,w2,w3,w4,w5^	2.3 ± 1.0	2.2 ± 1.0 ^w11^	5.7 ± 0.5 ^w1,w4,w5,w8,w10,w13,w16^	324.3 ± 71.9 ^w1,w2,w3,w4,w5^
W13	8.4 ± 0.6	9.8 ± 0.4 ^w2,w3,w4^	1.9 ± 0.7	1.8 ± 0.8	6.7 ± 0.6 ^w7,w12,w15,w16,w17^	387.7 ± 88.7 ^w4,w17^
W14	8.5 ± 0.6	9.8 ± 0.3 ^w1,w3,w4^	2.0 ± 0.7 ^w4^	1.9 ± 0.8	6.2 ± 0.7	368.5 ± 77.4 ^w2,w3,w4,w5,w17^
W15	8.4 ± 0.7	9.8 ± 0.4	2.0 ± 0.9	2.4 ± 1.1	6.0 ± 0.5 ^w1,w4,w5,w13^	386.0 ± 109.4 ^w4,w17^
W16	8.5 ± 0.6	9.7 ± 0.7	2.3 ± 0.7	2.3 ± 1.0 ^w11^	6.1 ± 0.5 ^w4,w12,w13^	366.1 ± 72.2 ^w2,w3,w4,w5,w17^
W17	8.3 ± 0.8	9.8 ± 0.6	2.0 ± 0.9	2.2 ± 1.1	5.4 ± 0.9 ^w1,w4,w5,w8,w13^	251.7 ± 61.0 ^w1–w5;w7-w11,w13–w16^

W: week; DOMS: delayed onset muscle soreness; A.U.: arbitrary units; RPE: Rate of perceived exertion measured in the CR-10 Borg’s scale; session-RPE: multiplication of time of session by the Borg’s scale score; ^w^: significant different at *p* < 0.05 in comparison to weeks 1^w1^, 2^w2^, 3^w3^, 4^w4^, 5^w5^, 6^w6^, 7^w7^,8^w8^, 9^w9^, 10^w10^, 11^w11^, 12^w12^, 13^w13^, 14^w14^, 15^w15^, 16^w16^, and 17^w17^.

**Table 4 ijerph-19-02935-t004:** Correlation coefficient (*r* and (95%confidence interval)) between well-being and training/match intensity outcomes of the same week.

	RPE	Session-RPE
Sleep	*r* = −0.018 (−0.109;0.074) *p* = 0.699	*r* = −0.119 * (−0.209;−0.028) *p* = 0.010
Readiness	*r* = −0.093 (−0.183;−0.002) *p* = 0.046	*r* = −0.235 ** (−0.319;−0.146) *p* < 0.001
Fatigue	*r* = 0.170 ** (0.080;0.258) *p* < 0.001	*r* = 0.325 ** (0.240;0.404) *p* < 0.001
DOMS	*r* = 0.111 * (0.071;0.249) *p* = 0.017	*r* = 0.161 ** (0.486;0.249) *p* < 0.001

DOMS: delayed onset muscle soreness; A.U.: arbitrary units; RPE: Rate of perceived exertion measured in the CR-10 Borg’s scale; session-RPE: multiplication of time of session by the Borg’s scale score; * significant at *p* < 0.05; ** significant at *p* < 0.01.

**Table 5 ijerph-19-02935-t005:** Correlation coefficient (*r*) between well-being of the previous week and training/match intensity outcomes of the following week.

	RPE	Session-RPE
Sleep	*r* = −0.091 (−0.184;0.004) *p* = 0.059	*r* = −0.091 (−0.184;0.003) *p* = 0.058
Readiness	*r* = −0.075 (−0.168;0.020) *p* = 0.122	*r* = −0.115 * (−0.207;−0.021) *p* = 0.017
Fatigue	*r* = 0.164 ** (0.070;0.254) *p* < 0.001	*r* = 0.262 ** (0.171;0.347) *p* < 0.001
DOMS	*r* = 0.102 * (0.004;0.191) *p* = 0.035	*r* = 0.099 * (0.497;0.626) *p* = 0.040

DOMS: delayed onset muscle soreness; A.U.: arbitrary units; RPE: Rate of perceived exertion measured in the CR-10 Borg’s scale; session-RPE: multiplication of time of session by the Borg’s scale score; * significant at *p* < 0.05; ** significant at *p* < 0.01.

**Table 6 ijerph-19-02935-t006:** Correlation coefficient (*r*) between training/match intensity of the previous week and well-being outcomes of the following week.

	Sleep	Readiness	Fatigue	DOMS
RPE	*r* = 0.049 (−0.046;0.142) *p* = 0.311	*r* = −0.135 ** (−0.227;−0.041) *p* = 0.005	*r* = 0.202 ** (0.109;0.290) *p* < 0.001	*r* = 0.122 * (0.028; 0.214) *p* = 0.011
Session-RPE	*r* = 0.021 (−0.074;0.115) *p* = 0.667	*r* = −0.167 ** (−0.257;−0.073) *p* < 0.001	*r* = 0.282 ** (0.193;0.367) *p* < 0.001	*r* = 0.134 * (0.040; 0.225) *p* = 0.005

DOMS: delayed onset muscle soreness; A.U.: arbitrary units; RPE: Rate of perceived exertion measured in the CR-10 Borg’s scale; session-RPE: multiplication of time of session by the Borg’s scale score; * significant at *p* < 0.05; ** significant at *p* < 0.01.

## Data Availability

Data available on request due to privacy. The data presented in this study are available on request from the first author.

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
