# Peer review of "Weekly Variations of Well-Being and Interactions with Training and Match Intensities: A Descriptive Case Study in Youth Male Soccer Players"

_ijerph, 2022, doi:10.3390/ijerph19052935_

Round 1

Reviewer 1 Report

The main aim of this study was two-fold: 1.) to analyze the weekly variations of well-being and training/match intensity measures in youth soccer players; and 2.) to test relations between well-being and training intensity outcomes. Regarding the authors, I would like to congratulate and thank them for their effort and motivation involved in this research study. The presentation of the research is well documented, with a scientific basis and respects the latest standards regarding the highest level scientific publications. The methodology was chosen correctly. The conclusions support and result from the research and open new directions for future research. The submitted work is interesting and essentially exhausts the subject under discussion.

However, I believe that the article would gain value if the topic of psychological predispositions to play football in the context of various stimuli and monitoring the fixation of results and techniques were expanded. There are many valuable publications on this subject that were not cited in the article. I especially recommend the authors to read the following publications:

a) Giovanni Esposito, Gaetano Raiola (2020). Monitoring the performance and technique consolidation in youth football players, "Trends in Sport Sciences", no. 27 (2), pp. 93-100.

b) Gert-Jan De Muynck, Sofie Morbée, Bart Soenens, Leen Haerens, Ona Vermeulen, Gert Vande Broek & Maarten Vansteenkiste (2021). Do both coaches and parents contribute to youth soccer players ’motivation and engagement? An examination of their unique (de) motivating roles, "International Journal of Sport and Exercise Psychology", no. 19 (5), pp. 761-779.

c) Rosendo Berengüí, Rafael Carralero, Maria A. Castejón, Juan A. Campos-Salinas, Cantón E. Values ​​(2021). Motivational orientation and team cohesion amongst youth soccer players. "International Journal of Sports Science & Coaching", November.

In summary, I consider the research carried out to be important, which should be published as a research article. With the above minor suggestions, the article will be suitable for publication in the International Journal of Environmental Research and Public Health.

Reviewer 2 Report

Manuscript ID: ijerph-1613597
Type of manuscript: Article
Title: Weekly variations of well-being and interactions with training and
match intensities: a descriptive case study in youth male soccer players

The work estimate perceptive sleep quality, readiness, fatigue and delayed onset muscle soreness using an adjusted version of the Hooper questionnaire, as well as, a rate of perceived exertion using the CR-10 Borg’s scale. The studies were performed with twenty-seven young soccer players during seventeen consecutive weeks with the 64 training sessions and 19 matches.

The obtained results were described in detail using advanced statistical analysis. Many of the results confirmed previous studies.

The work may be of interest to IJERPH readers.

I have one question to the work:

Is there a double dot in line 170?

Reviewer 3 Report

The present work approach relevant for professionals working with soccer: variations of well-being and training. The work presents good writing, as well as evaluated in the appropriate readability to test the proposed hypotheses. The work demonstrates the potential for publication, however, some important questions that may be influencing the findings of the study must be answered and I would also like to point out some suggestions to improve the quality of the manuscript:

- I recommend that the authors better describe how the training sessions are organized to understand the demand and the definition of the stimulus proposed by the coach.

- I suggest including a simple equation in the methods describing how the authors calculated the session-RPE to make it easier to differentiate from the RPE.

- I recommend the authors include a description table (baseline characteristics) of the participants, such as age, total body mass, height, BMI, fat percentage, absolute fat mass and absolute lean mass. I imagine they are variables already collected by the soccer team. This will help to understand more clearly the starting point of the research.

- I noticed a typo on page 5, lines 169/170 in the value of F which I imagine is correct “F=8.734” instead of “F=0.8.734”.

- I recommend authors revise figure 1 for separate figures for each parameter. Figure 1 has a lot of data and makes it difficult to understand the many statistical tests performed. In this case, I suggest you even point out the points observed with significance for better visualization for each parameter (both for the post-training comparison and for the weekly comparison). An additional suggestion would be to use charts with connected lines rather than bar charts, as it is a continuous analysis over time.

- In figure 1, please describe the meanings for all acronyms.

- Did the authors carry out any type of control on the athletes' current or weekly consumption or does the club have data collected by nutritionists/dietitians? Feeding is closely connected with the key parameters used for this investigation.

- Did the authors perform an analysis of intra-individual variation for the parameters? If so, it would be interesting to add an analysis of whether different positions behave differently for parameters throughout the season. In my opinion, the authors will not only find more data, but also a greater understanding of the reason for the observed variations.

Round 2

Reviewer 3 Report

After receiving the response to all questions and concerns from my point of view, I believe that the manuscript presents a higher quality for the reader to understand the significance of the research, as well as its limitations. The limitations do not affect the authors' conclusion, but they bring more maturity and caution to the reader who may be faced with these questions.